# Analysis of AUDIT Domains in Freshman Students in Spain: Three Cross-Sectional Surveys (2005, 2012 and 2016)

**DOI:** 10.3390/ijerph19137799

**Published:** 2022-06-25

**Authors:** Alicia Busto Miramontes, Lucía Moure-Rodríguez, Alba Regueira, Leonor Varela, Montserrat Corral, Adolfo Figueiras, Francisco Caamano-Isorna

**Affiliations:** 1Department of Public Health, University of Santiago de Compostela, 15782 Santiago de Compostela, Spain; lucia.moure.rodriguez@usc.es (L.M.-R.); alba.regueira@rai.usc.es (A.R.); leonor.varela@usc.es (L.V.); adolfo.figueiras@usc.es (A.F.); francisco.caamano@usc.es (F.C.-I.); 2Health Research Institute of Santiago de Compostela (IDIS), 15706 Santiago de Compostela, Spain; montse.corral@usc.es; 3Epidemiology and Public Health Networking Biomedical Research Centre (CIBERESP), 28029 Madrid, Spain; 4Department of Clinical Psychology and Psychobiology, University of Santiago de Compostela, 15782 Santiago de Compostela, Spain

**Keywords:** freshmen, audit domains, alcohol dependence, alcohol-related harm, cohorts

## Abstract

*Purpose*: We aimed to evaluate changes in the frequency of drinking, alcohol dependence and alcohol-related harm in freshman college students from 2005, 2012 and 2016, and identify risk-associated factors. *Method*: A cross-sectional study involving 5009 freshman students was carried out in Spain in 2005, 2012 and 2016. The Dimensions of Alcohol Use Disorders Identification Test (frequency of drinking, symptoms of dependence and alcohol-related harm) was analysed. Adjusted relative risks (RRs) and their 95% confidence intervals were estimated using negative binomial regression. *Results*: Place of residence, positive expectancies and early onset of alcohol consumption continue to be crucial conditions for developing patterns of risky consumption, alcohol dependence and harm. Women and men were more vulnerable to alcohol harm and dependence when living away from home, having higher economic status and positive alcohol expectancies. Cohorts from 2012 and 2016 were at lower risk of risky alcohol consumption. Men belonging to the 2012 and 2016 cohorts were at lower risk of alcohol negative consequences compared with the 2005 cohort. However, women remained vulnerable over time to alcohol dependence and harm. Starting drinking after 16 protects them both from alcohol dependence and harm. *Conclusions*: Women are more vulnerable to alcohol dependence and harm in recent cohorts of freshmen. Limiting access to alcohol at a younger age and working on false positive expectancies could benefit freshmen by avoiding alcohol damage and alcohol dependence.

## 1. Introduction

Problematic drinking continues to be an important concern for young adults, particularly college students. One of the groups more likely to engage in risky alcohol behaviours are freshmen because college is commonly a student’s first experience living away from home, and they can be overwhelmed by the need for social approval and the lifestyle changes that accompany independence [1,2,3].

Investigations on first-year college students [4] have shown a relationship between stress and alcohol consumption. Alcohol intake was related to tension-reduction drinking motives, gender, and behavioural under-control. Paradoxically, alcohol consumption increased psychological distress in these students [5].

College students, including freshmen, are also more vulnerable to suffering alcohol-related damage, being more prone to traffic accidents [6], alcohol-related injuries [7] and participation in unsafe sex [8]. Several investigations suggest differences in alcohol damage as well as alcohol dependence by gender and college years, some of them considering females more vulnerable to alcohol consequences than males and vice versa [9]. An investigation undertaken in Norway [10] showed that younger college students had higher rates of risky and harmful alcohol use compared with older ones. Nevertheless, the rates of dependent alcohol use were similar. They found that rates of risky, harmful and dependent alcohol use remained relatively stable over 8 years of follow-up.

Alcohol consumption during the first years of college has also been linked to negative behaviours over the years [11]. In fact, these initial experiences can predict future trajectories of alcohol consumption as well as predict college success and subsequent adult roles and independence [12,13].

Several studies on freshman students have discovered that the age of onset of alcohol consumption and the expectations about alcohol use are among the factors that most contribute to risky alcohol consumption, though living away from home and high maternal educational level also confer an increased risk [14,15,16].

The most prevalent drinking patterns (binge drinking and risky alcohol consumption) have been widely studied in college and freshman students [17,18]. Emerging research, such as that carried out by Mochrie et al. [19], suggests that beverage type preferences predict alcohol-related negative consequences such as dependence and harm among college students. Several studies carried out over the last 10 to 20 years have detected polarised alcohol consumption (abstainers and frequent binge drinkers increased over time) [20,21], affecting females more severely than males.

Our research group carried out a pooled analysis of three cross-sectional surveys investigating Spanish freshman students (2005, 2012 and 2016) and found an increase in prevalence rates of alcohol consumption among women during the study period (10 years), whilst finding no statistically significant difference in the prevalence rates among men [15].

The World Health Organization´s (WHO) AUDIT test [22], as is widely known, is the current gold standard screening instrument for the detection of hazardous and harmful drinking in the general population [22]. The AUDIT score was originally designed to cover three conceptual domains of hazardous alcohol use—risky alcohol consumption, symptoms of alcohol dependence and alcohol-related harm—and has been validated in numerous countries and subpopulations [23,24].

To our knowledge, only a few authors have studied the sociodemographic predictors of the AUDIT domains. Cook et al. [25] evidenced that the third domain (alcohol-related harm) was inversely related to educational level and socioeconomic status in a sample of Russian men. Another study carried out in Great Britain by Smith et al. [26] found that higher scores for the first dimension (risky alcohol consumption) were associated with male gender, younger age, lower educational level and anxiety disorders, as well as suicidal attempts. Higher punctuations on the third dimension (alcohol-related harm) were also associated with anxiety and depressive disorders, phobia and suicidal attempts. The evolution of AUDIT domains across generations of first-year university students has not been well investigated.

The goal of this study was to examine how the three dimensions of AUDIT perform and vary in freshman students from 2005, 2012 and 2016 cohorts, as well as their relationships to sociodemographic variables.

## 2. Materials and Methods

### 2.1. Design, Population and Sample

The present investigation includes datasets of freshmen from Santiago de Compostela, Spain, from three cohorts (2005, 2012 and 2016). The data were collected between September and February. We used cluster sampling to select the participants. At least one of the first-year classes was randomly selected from each of the university faculties or schools. The number of classes selected from each of the university faculties or schools was proportional to the number of students. All freshman students attending the classes on the day of the survey were invited to participate in the study (n_2005_ = 992, n_2012_ = 836 and n_2016_ = 1497 females and n_2005_ = 371, n_2012_ = 449 and n_2016_ = 864 males). To achieve more reliable results, the study was performed following the STROBE statement [27]. This study was approved by the Bioethics Committee of the Universidade de Santiago de Compostela. Subjects were informed both verbally and in written format that participation was voluntary, anonymous and that the possibility to opt out was available at any time.

### 2.2. Data Collection Procedures

Students attending selected first-year classrooms were invited to participate in the study. Alcohol use across the three domains was measured using the Galician validated version of the Alcohol Use Disorders Identification Test (AUDIT) [28]. In addition to the AUDIT, we used another questionnaire to determine factors potentially associated with alcohol use, such as parental educational level and their alcohol use, alcohol-related problems and age of onset of alcohol use.

This questionnaire also included a question that was specifically designed to measure alcohol-related expectancies. This question was generated using items from a previous questionnaire administered to a population of young Spanish adults [29]. The freshman students of 2005 and 2012 were asked to rank 14 expectancies about the effects of alcohol. Subsequently, we determined the expectancies ranked in the top seven positions by adding 1 point for each positive expectation and resting another point for each negative one as follows: it adds fun (+1), it helps me to socialise (+1), to feel more relaxed (+1), to forget about problems (+1), to endure problems (+1), it causes irritability (−1), anxiety (−1), depression (−1), confusion (−1), sleep-related problems (−1), nervousness (−1), aggression (−1), loss of control (−1), and heaviness/drowsiness (−1). As for participants in the 2016 study, expectancies were measured using the Alcohol Expectancy Questionnaire—Adolescent, Brief (AEQ—AB) [30].

### 2.3. Definition of Variables

#### 2.3.1. Independent Variables

Age of onset of alcohol consumption: In accordance with previous studies [14], the age of onset of alcohol use was categorised into four groups (after 16 years old, at 16, at 15, and before the age of 15).

Several sociodemographic variables were considered: gender, place of residence (parental home/away from the parental home), and maternal education level (primary school/high school/university).

Score of alcohol expectancies: The numbers of positive and negative alcohol-related expectancies for the 2005 and 2012 datasets were estimated as explained earlier. The score generated ranged from −7 to 5, −7 being the maximum number of negative and 5 being the maximum number of positive expectancies, respectively. For the 2016 study, a score was generated from the Alcohol Expectancy Questionnaire—Adolescent, Brief (AEQ—AB) [30]. For the analysis, all scores were divided into tertiles.

#### 2.3.2. Dependent Variables

AUDIT test: comprises the full original version of the AUDIT, which consist of 10 questions covering all three domains.

The first domain of the AUDIT test (risky alcohol consumption): the first three questions of the test: 1. “How often do you have a drink containing alcohol”? 2. “How many drinks containing alcohol do you have on a typical day when you are drinking”? 3. “How often do you have six or more drinks on one occasion”. The first domain accounts for 12 points [22].

The second domain of the AUDIT test (symptoms of alcohol dependence): 4th, 5th and 6th questions: 4. “How often during the last year have you found that you were not able to stop drinking once you had started”? 5. “How often during the last year have you failed to do what was normally expected from you because of drinking”? 6. “How often during the last year have you needed a first drink in the morning to get yourself going after a heavy drinking session? Never/less than once a month/at least once a month/at least once a week/daily or almost daily”. The second domain accounts for 12 points [22].

The third domain of the AUDIT test (alcohol-related harm): 7th, 8th, 9th and 10th questions: 7. “How often during the last year have you had a feeling of guilt or remorse after drinking”? 8. “How often during the last year have you been unable to remember what happened the night before because you had been drinking”? 9. “Have you or someone else been injured as a result of your drinking”? 10. “Has a relative or friend or a doctor or another health worker been concerned about your drinking or suggested you cut down”? Never/less than once a month/at least once a month/at least once a week/daily or almost daily”. The third domain accounts for 16 points [22].

### 2.4. Statistical Analysis

The baseline data from the three previous cohort studies were pooled into a single dataset and analysed using negative binomial regression to compute adjusted relative risks (RRs) and 95% confidence intervals (95%CI) for the three dimensions of the AUDIT test. Generalised negative binomial regression models were chosen for this analysis because they are more flexible than traditional models and, thus, permit the analysis of correlated data. The study period was introduced into the model as a random variable. In addition, a chi-square test was used to compare the differences between samples. Pseudo R2 by Nalgelkerke were calculated. Data were analysed using generalised negative binomial models in R Statistics software (The R Foundation for Statistical Computing c/o Institute for Statistics and Mathematics, Vienna, Austria) [31].

## 3. Results

Participation in the study was very high, reaching up to 99.0% of the students attending classes on the day of the survey. The characteristics of the sampled population from 2005, 2012 and 2016 are described in Table 1, categorised by gender. The average score of the global AUDIT test was 5.51 (P_25_:2; P_75_:8) for females and 6.93 (P_25_:3; P_75_:10) for males.

We can appreciate significant differences between cohorts according to the study year and gender. Thus, we can see that in the 2016 cohort, the age of onset of alcohol consumption is delayed from 15 to 16 years for both genders. We also observed a significant improvement in maternal educational level, as the number of women accessing university studies for both genders increased in the recent cohorts. For the AUDIT domains, the frequency of alcohol consumption decreased throughout the cohorts for both genders, as reflected by the mean of the first domain score, which varied from 3.80 in women and 5.04 in men to 3.28 and 4.04, respectively. As far as the second domain is concerned, students presented greater scores for dependency symptoms in 2016 cohorts than in previous ones, increasing from a mean of 0.446 in women and 0.70 in men in 2005 to 0.843 and 0.875 in 2016. The third domain also suffered variations, and the means of questions representing alcohol-related harm increased for both genders, changing from 1.18 to 1.65 in women and from 1.46 to 1.82 in men in the 2005 cohort and 2016 cohort, respectively. The differences related to the second and third domains are only significant for women.

Table 2, Table 3, Table 4 and Table 5 show the relationships between the global AUDIT test and the first, second and third domains of the AUDIT and the remaining sociodemographic variables. The main results of the analysis are presented below.

### 3.1. Age of Onset of Alcohol Use

Delaying the onset of alcohol consumption was a protection factor for alcohol’s negative consequences in both female and male freshman students. Those who started drinking after 16 were at lower risk than those who started drinking before they were 15 or younger. This tendency was observed for the general AUDIT as well as for the three domains (risky drinking, alcohol dependence and harm related) and for both genders. The RR of the global AUDIT were (RR = 0.44 (95%CI: 0.4–0.49)) for females and (RR = 0.44 (95%CI: 0.39–0.51)) for males. Delaying the first drink until the age of 16 years exerted a significant protective effect in relation to the second and third domains both in females (RR = 0.29 (95%CI: 0.23–0.36)) and males (RR = 0.33 (95%CI: 0.25–0.43)).

### 3.2. Place of Residence

The global AUDIT test score was significantly higher for male and female students living outside their parental homes (RR = 1.19 (95%CI: 1.12–1.26) and RR = 1.19 (95%CI: 1.10–1.28), respectively). The RR obtained for women when analysing alcohol dependence (second domain) was 1.34 (95%CI: 1.13–1.59). The tendency in males did not reach statistical significance. An additional risk was observed for both genders when analysing alcohol-related harm (third domain), as reflected by RR = 1.29 (95%CI: 1.14–1.45) for women and RR = 1.33 (95%CI: 1.14–1.55) for men.

### 3.3. Maternal Educational Level

For the global AUDIT test, we found significant differences in the maternal educational level of the freshmen from our three cohorts. In fact, female students whose mothers had completed secondary or university education had an additional risk of 7% and 8%, respectively, in comparison to the rest of the students (RR = 1.07 (95%CI: 1.01–1.13); RR = 1.08 (95%CI: 1.02–1.15)). This tendency remained stable when evaluating risky alcohol consumption (first AUDIT domain), in which students whose mothers had a university education also showed an increased risk of 6% of risky alcohol consumption (RR = 1.06 (95%CI: 1.01–1.11)).

In relation to the second AUDIT domain, the results revealed that the risk of alcohol dependence was 25% higher for female students whose mothers had attained a university degree (RR = 1.25 (95%CI: 1.06–1.47)). This tendency was not maintained when adjusted RRs were calculated.

For the third domain, alcohol-related harm, we evidenced that males whose mothers completed secondary school and females whose mothers held a college degree were also at risk of alcohol-related harm (RR = 1.21 (95%CI: 1.01–1.45) and RR = 1.14 (95%CI: 1.01–1.28), respectively).

### 3.4. Alcohol Expectancies

Greater alcohol expectancies were associated with a higher risk of alcohol consumption. This variable was the one that most influenced alcohol consumption among females and males in the three cohorts, and this relationship was observed for the global AUDIT score as well as for each of its three domains. For the second domain (alcohol dependence), higher RRs were found for both females (RR = 2.15 (95%CI: 1.81–2.55)) and males (RR = 2.23 (95%CI: 1.72–2.88)).

### 3.5. Period-Based Analysis of the Pooled Data of 2005, 2012 and 2016

When the total AUDIT score was calculated for each cohort, we found that students belonging to the 2012 and 2016 cohorts were protected in relation to the 2005 cohort. The RR observed in the 2016 cohort was 0.90 (95%CI: 0.85–0.96) in women and 0.76 (95%CI: 0.7–0.84) in men. This protective effect was also found for the 2012 cohort, in which women achieved a RR = 0.91 (95%CI: 0.85–0.98) and males a RR = 0.85 (95%CI: 0.77–0.94).

The analysis of the first domain of the AUDIT related to risky alcohol consumption showed that this protection remained stable for both cohorts. The RR was 0.76 (95%CI: 0.73–0.8) for females and 0.72 (95%CI: 0.67–0.77) for males in the 2016 cohort, and 0.85 (95%CI: 0.8–0.9) and 0.85 (95%CI: 0.79–0.92), respectively, for the 2012 cohort.

The results of the evaluation of the second AUDIT domain (signs of alcohol dependence) revealed that the risk of alcohol dependence in female students was higher in the 2016 cohort (RR = 1.52 (95%CI: 1.28–1.8)) in comparison to the 2005 cohort. On the contrary, male freshman students from the 2012 cohort showed lower risks of alcohol dependence (RR = 0.74 (95% CI: 0.55–1.00)) in relation to students from the 2016 cohort.

The 2016 female cohort also showed an increased risk of alcohol-related harm (third domain of AUDIT), obtaining a RR of 1.41 (95%CI: 1.25–1.58). The 2016 male cohort experienced a certain level of protection from suffering alcohol harm (RR = 0.79 (95%CI: 0.66–0.94)).

## 4. Discussion

In our study, living away from home, having positive alcohol expectancies and initiating alcohol consumption prematurely predispose freshman students to risky drinking patterns, alcohol dependence and alcohol-related harm. Female students appear to be more vulnerable to alcohol consequences, especially alcohol dependence and alcohol-related harm, when they live away from home, have a higher socioeconomic status determined by maternal educational level, maintain positive alcohol expectancies and have entered university in more recent years (2016 cohort). The risk of alcohol damage in male students increases when they live far away from home and have positive expectations of alcohol consumption.

We discovered that cohorts entering university in the last 10 years (2012 or 2016) were protected from alcohol consumption in comparison to their colleagues from 2005, and they also showed less risky alcohol consumption (first domain). A previous study carried out by our group [15] also evidenced a decreasing trend in the development of risky consumption patterns. Participants recruited in 2012 and 2016 faced lower risks in comparison to those from 2005. Nevertheless, in this aforementioned study, women from the 2016 cohort presented higher risks for the second and third AUDIT domains, which implies a higher risk of alcohol dependence and alcohol-related damage than their colleagues from earlier cohorts. On the contrary, a protective tendency was observed in men over time. This finding is consistent with other investigations, such as that carried out by Clarke et al. [9], suggesting that women are currently more prone to experience more severe alcohol problems at college than men. Our previous investigation is also concordant with these results, showing higher prevalence rates of alcohol consumption among women over freshmen, which may result in an increased likelihood of alcohol dependence and harm [15]. The results may also reflect the polarisation of drinking affecting female students more directly [20].

Another possible explanation for our findings could be related to the doubtful interpretation of the AUDIT domains, as Bernards et al. [32] have reported previously. It has been debated that the drinking frequency domain could contribute unequally to the total global score. This may lead to an inappropriate identification of some drinkers as hazardous drinkers. Due to this, our findings regarding the frequency of drinking and alcohol´s negative consequences must be interpreted with caution. Special attention should be paid to the third AUDIT question, which is related to binge drinking frequency, because it does not differentiate between gender.

The differences in alcohol dependence and harmful consequences of alcohol consumption between cohorts might be due to the social changes that occurred during this period. Several investigations suggest [33] that the 2012 and 2016 cohorts present an increased risk of harm and dependence despite lower alcohol intake in relation to their predecessors. Trend studies in the United States [34] have found that younger generations, especially those starting college after 2012 [35], were more affected by the rise in digital media and were more sensitive to mood disorders, suicidal thoughts, frustration and dependence, creating a possible cohort effect.

As far as maternal educational level is concerned, we found that students of both genders whose mothers had secondary and university degrees were at higher risk of suffering alcohol consequences. Female students whose mothers attended college also presented an increased risk of alcohol dependence and damage. The findings are consistent with previous research that found high maternal educational level is an indicator of student socioeconomic status, which conferred an increased risk for alcohol misuse [14]. This implies that medium–high maternal education level and possibly higher household incomes would increase the risk of alcohol-related harm. It cannot be disregarded that alcohol consumption might also have been influenced by the financial crisis that Spain suffered in 2008, as reflected by the fact that female students belonging to the 2012 cohort, whose mothers had a secondary education, achieved higher total AUDIT test scores.

Our results differ from those published by Cook et al. [25] and Smith et al. [26], who found that young males with lower economic status and lower educational levels had a higher risk of presenting with alcohol-related problems. This inconsistency could be due once more to the fact that frequency and volume questions might be less sensitive to socioeconomic variation in drinking behaviours than to questions about dependence and harm. Nevertheless, differences in our results may also originate from the substantial differences between populations (25–59-year-old Russian men [25] and the general population from Great Britain [26]). Unfortunately, the lack of studies on freshman students focused on AUDIT dimensions does not allow us to compare our results with those of other investigations.

Female and male freshman students living away from their parent´s home during the academic year achieved higher total AUDIT scores, as well as higher scores for each of the three domains separately. The risk varied during the studied periods, and greater RRs were found for females with respect to alcohol dependence (second dimension). Both genders were at risk of alcohol-related harm, although the risk was higher in males (third dimension) (29 and 33% increased risk, respectively).

Students of both genders with positive alcohol expectations were found to have higher risky consumption, alcohol dependence and alcohol-related harm. The risk doubled when students maintained their positive beliefs regarding alcohol. Our results are consistent with those found in previous studies in this cohort [4,5,11]. Students who started drinking in late adolescence attained lower total AUDIT scores, as well as lower scores for each of the dimensions: risky consumption, alcohol dependence and alcohol harm. A protective effect from alcohol dependence and alcohol damage was found when they started drinking after 16 years of age. Our results are consistent with the other scientific literature [36,37,38].

Our study presents three potential limitations: (1) The data were derived from self-administered questionnaires, which could result in an under- or overestimation of both independent and dependent variables [39]. However, we think that this is unlikely to occur when a validated test such as AUDIT is used. It has been widely proven that the AUDIT test performs well and produces reliable results in young adults and adolescent populations [40]. We consider that any misrepresentative data would probably affect descriptive data but not analytical findings [41]. (2) The use of a gender-specific instrument is recommended, although the validity of AUDIT for the Spanish college population has been proven [42,43]. (3). It cannot be dismissed that missing values for some independent variables could act as potentially confounding factors. We decided to focus on the most relevant factors to avoid extremely long questionnaires that could decrease the student participation rate.

## 5. Conclusions

Delaying the age of onset of alcohol use and intervening on false positive expectancies could help reduce harmful drinking in freshmen and, in turn, avoid alcohol damage and alcohol dependence. The risk of alcohol dependence and harm seems to be greater in female freshman students. Recent generations of male students seem to be protected from these effects in comparison to older cohorts. Further studies on freshman students are required to confirm these tendencies during the next few years.

## Figures and Tables

**Table 1 ijerph-19-07799-t001:** Sociodemographic characteristics and AUDIT test scores of 2005, 2012 and 2016 freshman cohorts by gender *.

	Female Cohort					Male Cohort				
	2005 n = 992	2012 n = 836	2016 n = 1497	*p*-Value *	All	2005 n = 371	2012 n = 449	2016 n = 864	*p*-Value *	All
**Age at onset of alcohol use (%)**										
<15 years old	19.0	17.4	4.9		12.2	18.1	16.9	6.0		11.6
15 years old	38.9	38.3	19.6		29.9	36.9	41.3	16.9		27.8
16 years old	25.6	31.5	61.3		43.4	21.6	26.6	61.9		43.6
>16 years old	16.5	12.8	1.41	<0.001	14.5	23.4	15.2	15.2	<0.001	17.0
**Residence (%)**										
In parental home	24.7	20.4	20.1		21.6	29.7	24.5	23.9		25.4
Out of parental home	75.3	79.6	79.9	0.005	78.4	70.3	75.5	76.1	0.036	74.6
**Maternal educational level (%)**										
Primary school	41.8	26.2	31.0		33.1	32.0	23.7	23.7		25.6
High school	33.6	34.9	26.4		30.7	27.6	30.0	30.0		26.1
University	24.6	38.9	42.5	<0.001	36.2	40.3	46.3	52.7	<0.001	48.2
**AUDIT: mean**	5.42	5.17	5.77	0.006	5.51	7.65	6.76	6.72	0.015	6.93
Mode	0.3	0	0		0	0	0	0		0
Range	25	29	29		29	29	32	31		32
Percentile 25/50/75	2/5/8	2/4/8	2/5/8		2/5/8	3/7/11	3/6/9	3/6/10		3/6/10
**First domain: mean**	3.80	3.27	3.28	<0.001	3.43	5.04	4.34	4.04	<0.001	4.34
Mode	3	2	3		3	4	4	4		4
Range	12	11	10		12	11	11	12		12
Percentile 25/50/75	2/4/5	2/3/5	2/3/5		2/3/5	3/5/7	2/4/6	2/4/6		2/4/6
**Second domain: mean**	0.446	0.461	0.843	<0.001	0.628	0.701	0.593	0.875	0.003	0.76
Mode	0	0	0		0	0	0	0		0
Range	7	6	9		9	7	9	11		11
Percentile 25/50/75	0/0/1	0/0/1	0/0/1		0/0/1	0/0/1	0/0/1	0/0/1		0/0/1
**Third domain: mean**	1.18	1.45	1.65		1.46	1.91	1.83	1.82		1.84
Mode	0	0	0		0	0	0	0		0
Range	12	13	14	<0.001	14	14	12	16	0.843	16
Percentile 25/50/75	0/0/2	0/1/2	0/1/2		0/1/2	0/1/3	0/1/3	0/1/2		0/1/2

* Chi-square.

**Table 2 ijerph-19-07799-t002:** Influence of explanatory factors on global AUDIT test score by gender. Negative binomial regression.

	Females		Males	
	RR (95%CI)	Adjusted RR *^,a^ (95%CI)	RR (95%CI)	Adjusted RR *^,b^ (95%CI)
**Age at onset of alcohol use**				
<15 years old	1	1	1	1
15 years old	0.83 (0.78–0.89)	0.86 (0.81–0.92)	0.75 (0.69–0.81)	0.82 (0.75–0.9)
16 years old	0.62 (0.58–0.66)	0.63 (0.58–0.68)	0.59 (0.53–0.65)	0.58 (0.52–0.64)
After 16 years old	0.43 (0.39–0.47)	0.44 (0.40–0.49)	0.43 (0.38–0.49)	0.44 (0.39–0.51)
**Residence**				
In parental home	1	1	1	1
Out of parental home	1.17 (1.09–1.25)	1.19 (1.12–1.26)	1.17 (1.07–1.29)	1.19 (1.1–1.28)
**Maternal educational level**				
Primary school	1	1	1	1
High school	1.1 (1.02–1.18)	1.07 (1.01–1.13)	1.1 (0.98–1.23)	1.1 (1–1.2)
University	1.1 (1.03–1.18)	1.08 (1.02–1.15)	1.06 (0.96–1.17)	1.07 (0.98–1.16)
**Alcohol expectations**				
1 tertile	1	1	1	1
2 tertile	1.43 (1.33–1.53)	1.29 (1.21–1.37)	1.42 (1.28–1.56)	1.18 (1.09–1.28)
3 tertile	1.79 (1.67–1.92)	1.5 (1.42–1.60)	1.8 (1.62–2)	1.53 (1.4–1.67)
**Cohort**				
2005	1	1	1	1
2012	0.96 (0.89–1.04)	0.91 (0.85–0.98)	0.89 (0.79–0.99)	0.85 (0.77–0.94)
2016	1.07 (1.0–1.14)	0.90 (0.85–0.96)	0.88 (0.8–0.98)	0.76 (0.7–0.84)

* Adjusted by all variables included in the column: ^a^ Pseudo R2 (%) = 19.94; ^b^ Pseudo R2 (%) = 21.35.

**Table 3 ijerph-19-07799-t003:** Influence of explanatory factors in first domain (risky alcohol consumption) of AUDIT test score by gender. Negative binomial regression.

	Females		Males	
	RR (95%CI)	Adjusted RR *^,a^ (95%CI)	RR (95%CI)	Adjusted RR *^,b^ (95%CI)
**Age at onset of alcohol use**				
<15 years old	1	1	1	1
15 years old	0.82 (0.78–0.86)	0.86 (0.83–0.93)	0.77 (0.72–0.82)	0.85 (0.8–0.92)
16 years old	0.66 (0.63–0.7)	0.66 (0.62–0.7)	0.66 (0.62–0.71)	0.66 (0.61–0.71)
After 16 years old	0.54 (0.5–0.59)	0.54 (0.5–0.59)	0.54 (0.49–0.59)	0.54 (0.48–0.6)
**Residence**				
In parental home	1	1	1	1
Out of parental home	1.11 (1.05–1.17)	1.13 (1.07–1.18)	1.1 (1.02–1.17)	1.11 (1.05–1.18)
**Maternal educational level**				
Primary school	1	1	1	1
High school	1.07 (1.01–1.12)	1.04 (0.99–1.09)	1.05 (0.97–1.13)	103 (0.96–1.1)
University	1.06 (1.01–1.12)	1.06 (1.01–1.11)	1.05 (0.98–1.11)	1.03 (0.97–1.1)
**Alcohol expectations**				
1 tertile	1	1	1	1
2 tertile	1.28 (0.93–1.04)	1.2 (1.14–1.26)	1.27 (1.18–1.36)	1.03 (1.08–1.24)
3 tertile	1.49 (1.41–1.57)	1.32 (1.25–1.39)	1.49 (1.38–1.61)	1.39 (1.3–1.49)
**Cohort**				
2005	1	1	1	1
2012	0.98(0.93–10.4)	0.85 (0.8–0.9)	0.96 (0.88–1.03)	0.85 (0.79–0.92)
2016	0.94 (0.89–0.98)	0.76(0.73–0.8)	0.85 (0.79–0.91)	0.72 (0.67–0.77)

* Adjusted by all variables included in the column: ^a^ Pseudo R2 (%) = 20.38; ^b^ Pseudo R2 (%) = 24.40.

**Table 4 ijerph-19-07799-t004:** Influence of explanatory factors in second domain (alcohol dependence) of AUDIT test score by gender. Negative binomial regression.

	Females		Males	
	RR (95%CI)	Adjusted RR *^,a^ (95%CI)	RR (95%CI)	Adjusted RR *^,b^ (95%CI)
**Age at onset of alcohol use**				
<15 years old	1	1	1	1
15 years old	0.95 (0.79–1.14)	0.88 (0.73–1.06)	0.64 (0.51–0.8)	0.7 (0.54–0.91)
16 years old	0.51 (0.41–0.62)	0.57 (0.46–0.7)	0.42 (0.32–0.55)	0.47 (0.35–0.63)
After 16 years old	0.25 (0.18–0.33)	0.3 (0.21–0.41)	0.22 (0.15–0.32)	0.3 (0.2–0.45)
**Residence**				
In parental home	1	1	1	1
Out of parental home	1.32 (1.11–1.56)	1.34 (1.13–1.59)	1.19 (0.95–1.47)	1.26 (1.00–1.58)
**Maternal educational level**				
Primary school	1	1	1	1
High school	1.1 (0.93–1.31)	1.12 (0.95–1.33)	1.18 (0.91–1.54)	1.26 (0.97–1.65)
University	1.25 (1.06–1.47)	1.17 (0.99–1.37)	1.14 (0.9–1.43)	1.13 (0.89–1.44)
**Alcohol expectations**				
1 tertile	1	1	1	1
2 tertile	1.73 (1.45–2.07)	1.54 (1.29–1.84)	1.76 (1.38–2.25)	1.39 (1.08–1.78)
3 tertile	2.46 (2.07–2.92)	2.15 (1.81–2.55)	2.76 (2.14–3.56)	2.23 (1.72–2.88)

* Adjusted by all variables included in the column: ^a^ Pseudo R2 (%) = 10.57; ^b^ Pseudo R2 (%) = 8.84.

**Table 5 ijerph-19-07799-t005:** Influence of explanatory factors in third domain (alcohol harm) of AUDIT test score by gender. Negative binomial regression.

	Females		Males	
	RR (95%CI)	Adjusted RR *^,a^ (95%CI)	RR (95%CI)	Adjusted RR *^,b^ (95%CI)
**Age at onset of alcohol use**				
<15 years old	1	1	1	1
15 years old	0.8 (0.7–0.91)	0.78 (0.68–0.9)	0.73 (0.61–0.86)	0.8 (0.66–0.96)
16 years old	0.57 (0.5–0.66)	0.59 (0.5–0.68)	0.5 (0.42–0.6)	0.47 (0.39–0.58)
After 16 years old	0.28 (0.23–0.34)	0.29 (0.23–0.36)	0.33 (0.26–0.43)	0.33 (0.25–0.43)
**Residence**				
In parental home	1	1	1	1
Out of parental home	1.3 (1.15–1.46)	1.29 (1.14–1.45)	1.33 (1.13–1.55)	1.33 (1.14–1.55)
**Maternal educational level**				
Primary school	1	1	1	1
High school	1.15 (1.02–1.3)	1.11 (0.98–1.25)	1.2 (0.99–1.44)	1.21 (1.01–1.45)
University	1.2 (1.07–1.36)	1.14 (1.01–1.28)	1.11 (0.94–1.31)	1.11 (0.95–1.31)
**Alcohol expectations**				
1 tertile	1	1	1	1
2 tertile	1.56 (1.38–1.78)	1.42 (1.26–1.61)	1.47 (1.24–1.74)	1.2 (1.02–1.42)
3 tertile	2.02 (1.79–2.29)	1.76 (1.56–1.98)	1.9 (1.59–2.27)	1.59 (1.33–1.89)
**Cohort**				
2005	1	1	1	1
2012	1.24 (1.08–1.41)	1.24 (1.08–1.43)	0.96 (0.79–1.16)	0.93 (0.76–1.13)
2016	1.41 (1.25–1.58)	1.21 (1.07–1.37)	0.97 (0.82–1.14)	0.79 (0.66–0.94)

* Adjusted by all variables included in the column: ^a^ Pseudo R2 (%) = 10.95; ^b^ Pseudo R2 (%) = 10.67.

## Data Availability

All relevant data are included within the paper. Data contain potentially identifying information and sensitive participant information. For all these reasons, the authors must not upload the dataset to a stable, public repository. However, the authors agree to make freely available any materials and data described in the publication upon reasonable request to fran-cisco.caamano@usc.es, the principal investigator of this project and professor at the University of Santiago de Compostela.

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
