# Peer review of "Analysis of AUDIT Domains in Freshman Students in Spain: Three Cross-Sectional Surveys (2005, 2012 and 2016)"

_ijerph, 2022, doi:10.3390/ijerph19137799_

Round 1
Reviewer 1 Report
The following study reported on pooled analysis of three cross-sectional surveys investigating Spanish freshman students to determine alcohol consumption patterns, socio-demographic factors associated with negative effects of alcohol use disorder.
Introduction:
Several errors related to grammar, syntax, and word order are apparent in the introduction and throughout the manuscript. Please see the following examples lines: 48 and 58.
Are shots a specific type of alcohol beverage? The sentence (lines 65-66) is false, and the syntax needs to be improved.
The introduction contains three different versions for defining the aim or purpose of the manuscript. Please consider consolidating aims and generate a concise and consistent objective statement.
Line:76; as currently written, it remains unclear how the manuscript can provide "more precise estimates". If these data are estimates, they are not precise.
Methods:
Consider revising lines" 105-106. "each university faculty or school" is not clear.
Several grammar, syntax, and incorrect word usage identified.
Results:
Table 1. - consider labeling the three cohorts (2005, 2012, 2016) at the top of the table near female/male distinction.
Table legends/descriptions are insufficient and do not provide adequate information to interpret the information located in the tables. As currently written, it remains unclear what statistical comparisons are made in the tables and what analysis was used to justify the comparison.
Discussion:
As currently written, it remains unclear if several intext citations are missing. Rather than a citation, the manuscript contains "Error! Bookmark not defined".
The discussion should be revised to correct grammar, syntax, and word usage errors.
Consider describing more fully the data set for the 2016 cohort in comparison to the other cohorts. Important differences in this cohort are briefly and inadequately discussed.
The study is relevant and a valuable contribution to the global interest in AUD research; however, significant revisions are recommended.
Author Response
"Please see the attachment"

Reviewer 2 Report
Many thanksfor your paper.
IIt is interesting topic for me. I think that some reviews are need to improve the quality of the study.
In methods paragraph, please report a statement followed to perform the study, for example I suggest STROBE for observation studies. It will be cited opportunly.
Please report in the statistical methods, the statistical tests, or other analysis present in the tables. The significant level set and the software used too.
In Results you have to include the notes near the tables that explain the statistical tests or other information to support the readers.
In discussion empathize the limits of the studies.
Author Response
"Please see the attachment"

Reviewer 3 Report
This is an interesting study using three years of data to examine changes in drinking frequency, alcohol dependence, and alcohol-related harms among freshmen students. However, I have several questions that would strengthen the manuscript. First, it is necessary to emphasize why this research question needs to be further studied. Second, to verify the relationship between the independent variable and outcome variables, additional consideration of potential confounders should be required. Third, the authors need to revise the results more completely. For example, Table 2 and Table 3 have different titles, but have the same picture in both tables.
Author Response
"Please see the attachment"

Round 2
Reviewer 1 Report
The revisions have improved the quality of the manuscript. Please consider a final check of spelling and grammar.
Author Response
"Please see the attachment"

Reviewer 2 Report
Many thanks for your review. I have some others comments in consideration of the your STROBE application.
- The statistical method subparagraph could be improved. For example you haven't reported the significant level of your analysis. The test used for univariate analysis are lacked and the indexes used for description of the data too. See STROBE statement raccomandations.
- In tables 2,3,4 and 5 please clarify the adjustment of the RR (for example by age etc) and add the crude in the title for the RR not adjusted.
- It could be usefull to indicate in bold the RRs significant or other signifincat estimate.
- I would ask if it is possible to evaluate and report in tables the quality of the negative binomial regression model fit.
Author Response
"Please see the attachment"

Reviewer 3 Report
This paper is valuable in that it collected longitudinal data from college students and analyzed their drinking pattern changes.
- However, to increase the readability of readers, I would like to highly recommend the authors to receive research editing service.
- In addition, it is necessary to organize the manuscript systematically so that readers can easily follow the flow of manuscript.
e.g., presenting the rationales for setting the drinking start age as 15 to 16 years old; in the method section, presenting the fact that the maternal education level is used among the parental education levels
- I wonder why the authors did not present the alcohol expectation level by gender.
Author Response
"Please see the attachment"
